# Calcium-dependent pathway as a primary cause of hypoxic RGC damage in monkey retinal explants

**Emi Nakajima[1,2], Momoko Otsugu-Kobayashi[3], Takatoshi Uchida[1,2], Kana Orihara[1], Thomas R. Shearer[2†], Mitsuyoshi Azuma⬡[1,2,3]***

**1** Senju Laboratory of Ocular Sciences, Senju Pharmaceutical Corporation Limited, Portland, Oregon, United States of America, **2** Department of Integrative Biomedical & Diagnostic Sciences, Oregon Health & Science University, Portland, Oregon, United States of America, **3** Senju Laboratory of Ocular Sciences, Senju Pharmaceutical Corporation Limited, Kobe, Japan

† Deceased Dec. 4, 2024
* azumam@ohsu.edu

## Abstract

### Purpose

Retinal ganglion cells (RGCs) loss or degeneration in the retina is a hallmark of many sight-threatening diseases, including glaucoma and retinopathy. In our previous studies, calcium-activated cysteine protease calpain induced RGC damage under hypoxia/reoxygenation in the monkey retina explants, and the calpain inhibitor SNJ-1945 partially inhibited RGC damage. Calcium-independent proteases such as cathepsins exist in the retina, although the involvement of cathepsins in hypoxia-induced RGC damage is unclear. The purpose of the present study is to determine if cathepsins are involved in RGC damage during hypoxia/reoxygenation and to elucidate the role of calcium.

### Methods

The cathepsin inhibitors (Odanacatib and SID26681509) were tested for their specificity against three cathepsins in vitro. Calpain inhibitors (SNJ-1945, PD-151746, ABT-957, and C2I) were tested for their isozyme specificity in vitro. Monkey retinal explants were cultured under hypoxic conditions with 0.3% oxygen in the chamber, followed by reoxygenation. The calpain- or cathepsin-specific inhibitors were added to the hypoxic culture medium. BAPTA and BAPT-AM were also used to determine the calcium requirement in RGC damage. After the cultured period, retinal explants were prepared for flat mounts and retinal lysates. The flat mounted retinas were stained with propidium Iodide (PI) to assess RGC damage and with an antibody specific for calpain-specific α-spectrin break down product 150 kDa (SBDP150). Immunoblotting assays were performed for α-spectrin and calpains.

**Data availability statement:** All relevant data are within the manuscript.

**Funding:** This research was funded by Senju Pharmaceutical Co., Ltd. in Japan. The monkey tissue distribution program was supported in part by National Institutes of Health Grant P51OD011092 to the Oregon National Primate Research Center.

**Competing interests:** I have read the journal's policy and the authors of this manuscript have the following competing interests: Dr. Shearer has significant financial interests (research contract and/or consulting fee) in Senju Pharmaceutical Co., Ltd., and Drs. Azuma, Nakajima and Uchida are employees of Senju Pharmaceutical Co., Ltd. a company that may have a commercial interest in the results of this research and technology. These potential conflicts of interest have been reviewed and managed by OHSU Conflict of Interest in Research Committee. The authors would like to declare the following patents/patent applications associated with this research: the patent number for SNJ-1945 is WO2005056519. This does not alter our adherence to PLOS ONE policies on sharing data and materials.

## Results

The cathepsin inhibitors were confirmed to be specific to cathepsin without calpain inhibitory effects in vitro. The number of PI-positive RGCs increased in the hypoxic monkey retina; however, cathepsin inhibitors did not mitigate RGC damage. In contrast, RGC damage was completely inhibited by BAPTA and partially by BAPTA-AM. In hypoxic retinas, calpain-specific SBDP150 increased in the nerve fiber layer (NFL). Immunoblotting revealed an increase in SBDP150 and the activation of calpain 1. These changes were inhibited by BAPTA or BAPTA-AM, with BAPTA demonstrating a stronger effect compared to BAPTA-AM. Calpain inhibitors demonstrated lower specificity in vitro than previously reported. In hypoxic retina, each calpain inhibitor alleviated RGC damage and reduced SBDP150-positive staining in NFL. Furthermore, calpain inhibitors attenuated the activation of calpain 1 and the breakdown of α-spectrin, as assessed by immunoblotting.

## Conclusions

Calcium-independent cathepsins do not contribute to RGC damage in monkey retinal explants cultured under hypoxia/reperfusion. In contrast, calcium influx from extracellular sources plays a critical role in inducing RGC damage. Elevated intracellular calcium levels could activate calpains, leading to RGC damage; however, other calcium-dependent pathways might also be involved in this process.

## Introduction

The retina is one of the most metabolically active tissues, consuming oxygen more rapidly than other tissues including the brain [1]. Retinal hypoxia is an underlying mechanism in several sight-threatening disorders, including central retinal occlusion, ischemic central retinal vein thrombosis, complications of diabetic eye diseases, some types of glaucoma, and age-related macular degeneration. Among the various types of neuronal cells in the retina, retinal ganglion cells (RGCs) are the most vulnerable. RGC death leads to gradual vision loss and eventual blindness, as RGCs are responsible for transmitting visual signal to the brain [2,3]. Hypoxia can trigger the formation of mitochondrial reactive oxygen species (ROS), causing an increase in cytosolic calcium levels in the ischemic brain. Calcium is translocated from the extracellular space and/or organelles into intracellular spaces. The increase in cytosolic calcium can trigger cell death and promote the execution of cell death pathway [4,5].

Calpains (EC 3.4.22.52/53) comprise a family of 15 non-lysosomal, calcium activated, cysteine proteases, classified as either ubiquitous or tissue specific. Calpains play important roles in physiological and pathological processes, regulating signal transduction, cell proliferation, cell cycle progression, differentiation, apoptosis, membrane fusion, processing of arrestin, and platelet activation [6,7]. Conventional calpains 1 and 2 are particularly well studied in diseases. Excitotoxicity, anoxia, or ischemia increase the calcium concentration in cells to promote abnormal calpain

activation. Activated calpain cleaves several essential cytoskeletal proteins in neuronal axons and contributes to neuro-degenerative cell death. Calpain exists throughout the retina layers and in the cytoplasm and nucleus of retinal ganglion cells. Our previous studies with human and monkey retinal explants treated with hypoxia show that 1) calpains 1 and 2 were activated, 2) calpain-specific α-spectrin breakdown products (SBDPs) increased, and 3) calpain inhibitor SNJ-1945 partially inhibited RGC damage [8,9].

Calcium-independent cysteine proteases such as caspases and cathepsins also exist in eye tissues. Fourteen caspases are identified and classified by their known roles in apoptosis and inflammation. Apoptosis-related caspases are subclassified as initiator or executioner caspases, which organize cascade systems into intracellular apoptotic signals. In addition, caspases play a role in non-apoptotic cell death such as necroptosis, pyroptosis, and autophagy. However, our previous studies showed that caspase-3 did not contribute to RGC damage induced by hypoxia/reoxygenation in human retinal explant culture [9].

Cathepsins are another family of calcium-independent, cysteine proteases comprised of 11 enzymes B, C, F, H, K, L, O, S, V, X, and W. Cathepsins are generally located in the lysosomes of all organisms and play a key role in cell protein turnover [10,11]. Cathepsins are known to cause cell death in apoptosis, necrosis, autophagy, pyroptosis, and ferropto-sis by interacting with proteins. However, the involvement of cathepsins in hypoxia-induced RGC death is not clear. The purpose of the present study was to determine if calcium-independent cathepsins are involved in RGC damage caused by hypoxia in cultured monkey retinal explants. Since calpain inhibitor SNJ-1945 was partially effective in previous studies, several calpain inhibitors and calcium chelators were also tested to elucidate the role of calcium.

## Materials and methods

### Purification of porcine calpain-2

The porcine kidney was obtained from DART Co., Ltd. as part of a program aimed at the effective utilization of livestock by-products in Hokkaido, Japan. The procurement process followed ethical guidelines from Livestock Industry Bureau MAFF-J (ministry of agriculture, forestry and fisheries of Japan) for farm animal use, ensuring compliance with relevant regulations. Two porcine kidneys were rapidly frozen after removal, trimmed free of excess tissue or renal pelvis, and cut into small pieces (approximately 2 × 2 cm) before they thawed. About 250 g of these pieces were homogenized in 1200 mL of ice-cold 20 mM Tris-HCl (pH 8.0) with 5 mM ethylenediaminetetraacetic acid (EDTA), 0.1% 2-mercaptoethanol (2-ME), 0.1 mg/mL Pefabloc SC (MilliporeSigma, Burlington, MA, USA), 40 µg/mL bestatin, and 4 µg/mL pepstatin in a Waring blender HGBSS (Waring; Stamford, CT, USA) at 19,000 rpm using several bursts of 10 sec each with 20 sec cool-ing periods between each burst. The homogenate was centrifuged at 25,000 × g for 15 min at 4°C, and the supernatant was filtered through glass wool and a 5 µm filter (Merck) to remove floating lipids. The purification procedure was per-formed according to the method of Thompson and Goll [12] with some minor modifications: the DEAE-cellulose and the DEAE-TSK ToyoPearl 650S columns were substituted with DEAE sepharose fast flow (Merck) and TSKgel DEAE-5PW (30) (Tosoh; Tokyo, Japan) columns, respectively. After purification, calpain-2 was dialyzed, and glycerol was added for freeze-thaw protection. The final buffer formulation was 20 mM imidazole, 5 mM 2-ME, 1 mM EDTA, 1 mM EGTA, and 20% glycerol.

### In vitro inhibitory assay of calpains and cathepsins

Cathepsins K, B, and L were selected because of their known roles in cell death and early release following lysosomal membrane permeabilization [13]. The inhibitory activities of cathepsin inhibitors were assessed in vitro using human recombinant cathepsin K (MilliporeSigma), human liver cathepsin B (MilliporeSigma), and human liver cathepsin L (MilliporeSigma). Cathepsins and the cathepsin substrate, Z-Phe-Arg-AMC (PEPTIDE INSTITUTE, INC.), were dissolved in 50 mM sodium acetate buffer (pH 5.5) containing 2 mM dithiothreitol and 2 mM EDTA. A cathepsin K inhibitor, Odanacatib

(Selleck Chemicals, Houston, TX, USA) [14], and a cathepsin L inhibitor, SID26681509 (Bio-Techne Corporation) [15], were dissolved in dimethyl sulfoxide (DMSO) (Nacalai Tesque, Kyoto, Japan) before use in the assay. The final concentration of DMSO in the reaction solution was 2%. The cathepsin substrate, 20 µM Z-Phe-Arg-AMC, was incubated for 15 min with 20 ng/mL cathepsin K, 15 ng/mL cathepsin B, or 50 ng/mL cathepsin L, with or without each inhibitor at a final concentration of 0.001 nM to 100 µM in 100 µL of mixed reaction solution in a 96-well plate at 37°C. The fluorescence was measured using a microplate reader (GloMax Discover System; Promega). IC50 values from two independent experiments were averaged.

For measurement of inhibitory activities against calpains in vitro, Calpain-Glo Protease Assay (Promega; Madison, WI, USA) was performed using human erythrocyte calpain 1 (MilliporeSigma) and purified porcine calpain 2. Briefly, 24 µL of each calpain diluted with 20 mM Tris-HCl (pH 7.4), 1 µL of each inhibitor, and 25 µL of reaction buffer including $CaCl_2$ in a 96-well plate was incubated for 10 min at room temperature. The final concentration of $CaCl_2$ in the reaction solution was 5 mM. The luminescence was measured using a microplate reader (GloMax Discover System; Promega). IC50 values from two independent experiments were averaged.

SNJ-1945 (Senju Pharmaceutical), PD-151746 (Abcam, Cambridge, UK), C2I (prepared in-house), and ABT-957 (prepared in-house) were used as calpain inhibitors. PD-151746 as a selective calpain 1 inhibitor directed towards calcium binding sites of calpain [16], C2I as a relatively selective calpain 2 inhibitor contains a peptidyl α-ketoamide that binds to the catalytic triad of the calpain protease core [17], and ABT-957 as another α-ketoamide based calpain inhibitor [18].

## Monkey eyes

Globes were obtained from rhesus macaques (Macaca mulatta) at 1–7 years of age from the Oregon National Primate Research Center (Beaverton, OR, USA). The wide range of ages was unavoidable because they became available from experiments that were unrelated to the present studies. These eyes were acceptable for the purpose of our experiments. The eyes were dissected within 5 hr after death, and excised eyes were soaked in cold Hanks' balanced salt solution (HBSS; Corning, NY, USA) or Dulbecco's phosphate-buffered saline (DPBS; Corning). Experimental animals were handled in accordance with the Research in Vision and Ophthalmology Statement for the Use of Animals in Ophthalmic and Vision Research and with the Guiding Principles in the Care and Use of Animals (Department of Health Education and Welfare Publication National Institutes of Health 80−23).

## Retina explant culture

Retina explant culture was performed as described in our previous report [8], except for the induction of hypoxia. Briefly, a hypoxic chamber (Coy Laboratory Products, Grass Lake, MI, USA) was controlled at 0.3% oxygen with a 95% $N_2$/5% $CO_2$ gas mixture. Retinas were dissected into fan-shaped explants with 6 or 7 petals (> 5 mm) in HBSS. Each explant was placed with the RGC side facing up on a Millicell organotypic standing insert (0.4 µm, 30 mm diameter, MilliporeSigma) in six-well, culture plates. They were cultured for 2 hr at 37°C under 95% $N_2$/5% $CO_2$ in Neurobasal-A medium supplemented with B27, N2 supplements, 2 mM L-glutamine (Thermo Fisher Scientific, Waltham, MA, USA), and 100 µg/mL primocin (Invivogen, San Diego, CA, USA). Retinal explants were then chamber-cultured under hypoxic conditions for 16 hr in a culture medium with 0.5 mM glucose, under 0.3% oxygen. The retinas were then re-oxygenated for 8 hr in a culture medium with 5.5 mM glucose. Normoxic control retinas were cultured for 24 hr in a culture medium with 5.5 mM glucose. When present, calpain inhibitors (SNJ-1945, PD-151746, C2I, and ABT-957) and cathepsin inhibitors (Odanacatib and SID26681509) in 0.1% DMSO were used at the final concentration of 10 or 100 µM in the medium. BAPTA (Thermo Fisher Scientific) and BAPTA-AM (Thermo Fisher Scientific) were also used at a final concentration of 3 or 10 mM or a final concentration of 100 or 500 µM, respectively.

All inhibitors and BAPTA were treated simultaneously with hypoxia induction, and BAPTA-AM treatment was 2 hr before hypoxia induction.

## Staining of flat-mounted retinal explants

Formalin-fixed retinal explants were incubated overnight at −30°C in 1: 4 dilutions of DMSO-ethanol to promote antibody penetration. The explants underwent two cycles of freezing and thawing between −80°C and room temperature in 100% ethanol for 20 min. The retinas were then rehydrated in 70%, 50%, and 15% ethanol and PBS for 20 min each, followed by overnight incubation with 0.2% Triton X-100. The samples were stained with antibodies specific for calpain-specific SBDP150 [8] (1:100 dilution, Senju Pharmaceutical) and β-III tubulin (a neuron marker, 1:500 dilution, Merck). The nerve fiber layer (NFL) was observed approximately 5 mm from the tips of the fan-shaped retinal flat mounts. Confocal images were taken using a model LSM 880 spectral confocal system with a 20x 0.8 NA lens (Carl Zeiss, Oberkochen, Germany).

## Protein extraction from retinal explants and immunoblotting

Each retinal explant was sonicated in RIPA Lysis and Extraction Buffer (Thermo Fisher Scientific), with protease inhibitors cocktail (Complete Mini-EDTA-free; MilliporeSigma). Protein concentrations in lysates were measured by BCA assay (Pierce BCA Protein Assay Kit, Thermo Fisher Scientific) using bovine serum albumin as a standard. For immunoblotting, 5 µg of each sample was loaded and run on 4% to 12% gradient or 10% SDS-PAGE gels (NuPAGE Bis-Tris Protein Gels, Thermo Fisher Scientific) with 2-(N-morpholino)ethanesulfonic acid buffer or 3-(N-morpholino) propanesulfonic acid buffer (Thermo Fisher Scientific). Proteins were then electrotransferred to a polyvinylidene fluoride membrane (MilliporeSigma) at 100 V for 60 min or 1.5 hr. Membranes were then blocked with 5% skim milk in Tris-buffer saline (Bio-Rad Laboratories, Hercules, CA, USA) containing 0.05% Tween 20. Each membrane was probed with primary antibodies against α-spectrin (1:2000 dilution, Enzo Life Sciences, Farmingdale, NY, USA), SBDP150 (1:1000 dilution, Senju Pharmaceutical), calpain 1 (1:1000 dilution, Thermo Fisher Scientific), calpain 2 (1:500, GeneTex, Irvine, CA, USA), cathepsin K (1:500 dilution, Abcam), and β-actin (1:1000 dilution, Merck). For calpain 1, immunoreactivity was visualized using a secondary antibody conjugated to alkaline phosphatase and the AP Conjugate Substrate Kit (Bio-Rad Laboratories). For other proteins, immunoreactivities were visualized using secondary antibodies conjugated with rabbit (Santa Cruz Biotechnology) or mouse horseradish peroxidase enzyme (Cytiva, Wilmington, DE, USA) and ECL Plus detection reagents (Cytiva). Images of the membranes were captured with FluorChem FC2 imager (Cell Bio-sciences, Santa Clara, CA, USA). The conversion of pro-cathepsin K (38 kDa) to its mature form (27 kDa) was used as an indicator of cathepsin K activation.

## PI staining and RGC counting

To assess the disruption of membrane barrier function, propidium iodide (PI) was utilized [8]. Following hypoxia/reoxygenation treatment, retinal explants were incubated for 20 min in a medium containing a final concentration of 2 µg/ml of PI. For quantitative analysis, the RGC marker, RNA-binding protein with multiple splicing (RBPMS) [19] was used at a 1:100 dilution (PhosphoSolutions, Aurora, CO, USA). RBPMS-positive and PI-positive cells in the ganglion cell layer were counted in 2 images from 2 square areas (775 × 775 µm each) located 3–5 mm and 5–7 mm away from the tips of the fan-shaped retinal flat mounts. The percentage of PI-positive RGCs was calculated as [the number of PI and RBPMS-double positive cells/the number of all RBPMS-positive cells] × 100.

## In vitro inhibitory assay of calpains from retinal lysate

To confirm the isozyme specificities of calpain inhibitors in vitro, non-cultured monkey retinas were sonicated in a buffer containing 20 mM Tris (pH 7.5), 5 mM EGTA, 5 mM EDTA, 2 mM DTE. The lysate was then incubated for 2 hr with 10 mM $CaCl_2$ and with or without each calpain inhibitors SNJ-1945, PD-151746, C2I, or ABT-957 at a final concentration of 1, 10 and 100 µM at 37°C. Proteolysis was terminated by the addition of EGTA at a final concentration of 10 mM, and immuno-blotting was performed as described above.

## Statistical analysis

At least three independent experiments from different monkey eyes were conducted for all the studies in this report. Statistical analyses were performed by Dunnett's test (JMP17.0 statistical software; SAS Institute Inc., Cary, NC, USA). $P < 0.05$ was considered statistically significant.

## Results

### Cathepsins were not major contributors to RGC damage in hypoxic retina

When tested in vitro, Odanacatib showed strong inhibition of cathepsin K with an IC50 of 0.28 nM and moderate inhibition of cathepsin B with an IC50 of 167 nM. SID26681509 exhibited an IC50 of 60 nM for cathepsin L (Fig 1A). Neither inhibitor affected calpains 1 or 2, confirming the specificity of these cathepsin inhibitors without appreciable inhibition of the other cysteine proteases, calpains 1 and 2.

Monkey retinal explants treated with hypoxia/reoxygenation showed an increase in PI-positive RGCs, reaching 45–55%, compared to only 1% in normoxic retinal explants (Figs 1B & C). Treatment with Odanacatib (Fig 1B) or SID26681509 (Fig 1C) did not reduce the number of PI-positive RGCs induced by hypoxia/reoxygenation, suggesting that cathepsins K, B, and L are not involved in hypoxic retinal cell damage. In contrast, SNJ-1945 significantly decreased the number of PI-positive RGCs (Figs 1B & C), consistent with the previous report [20].

The representative immunohistochemical staining revealed an increase in calpain-specific SBDP150-positive staining in retinal fibers under hypoxia/reoxygenation conditions. Neither Odanacatib nor SID26681509 mitigated the increase in calpain-specific SBDP150-positive staining (Fig 1D). The immunoblot analysis revealed an increase in degraded α-spectrin at 145 and 150 kDa, while the caspase-3 mediated degradation of α-spectrin at 120 kDa was not detected (Fig 1E). Autolytic N-terminal truncations of calpain 1, including the partially activated form at 78 kDa and the fully activated form at 76 kDa [21], were observed in hypoxic retina, along with an increase in calpain-specific SBDP150. The cathepsin inhibitors did not prevent the degradation of α-spectrin, the increase in calpain-specific SBDP150, or the autolysis of calpain 1 induced by hypoxia/reoxygenation (Fig 1E). These results contrast with the calpain inhibitor SNJ-1945, which prevented these changes. The active mature form of cathepsin K was not induced in the hypoxic retina, as demonstrated by immunoblot analysis, although the inactive pro-form of cathepsin K was present (Fig 1F). In summary, all these observations support the conclusion that it was not cathepsins K, B, or L, but rather calcium-dependent calpain, that caused RGC damage following hypoxia/reoxygenation.

### Calcium ions were essential for hypoxia-induced RGC damage

Since classical calpains, such as calpains 1 and 2, require calcium ions for their activation [7], hypoxic monkey retinas were cultured with BAPTA to chelate extracellular calcium or BAPTA-AM to chelate intracellular calcium. Damaged PI-positive RGCs significantly increased to 60% in monkey retinal explants cultured under hypoxia/reoxygenation compared to 10% in normonic retinal explants (Figs 2A yellow arrows and 2B). At higher concentration, BAPTA almost completely inhibited RGCs damage, while the effect of BAPTA-AM was partial (Figs 2A yellow arrows and 2B).

Calpain-specific SBDP150 staining (red) was increased in the NFL in hypoxic retinal explants compared to the normoxic retina, which showed negligible SBDP150 signal (Fig 2C). BAPTA, even at a lower concentration, reduced SBDP150 staining induced by hypoxia, resembling the staining observed in normoxic retina. This effect was more pronounced compared to that of SNJ-1945. Higher concentrations of BAPTA-AM (500 µM) also inhibited the increase in SBDP150 staining (Fig 2C).

Immunoblotting further confirmed the increase in calpain specific SBDP150 in hypoxic retina and the inhibition of calpain-specific SBDP150 by BAPTA or SNJ-1945 (Fig 2D α-spectrin and SBDP150). Calpain 1 was autolyzed to a partially activated form at 78 kDa and a fully activated form at 76 kDa under hypoxia/reoxygenation (Fig 2D Calpain 1). Both BAPTA and SNJ-1945 inhibited the autolytic activation of calpain 1.

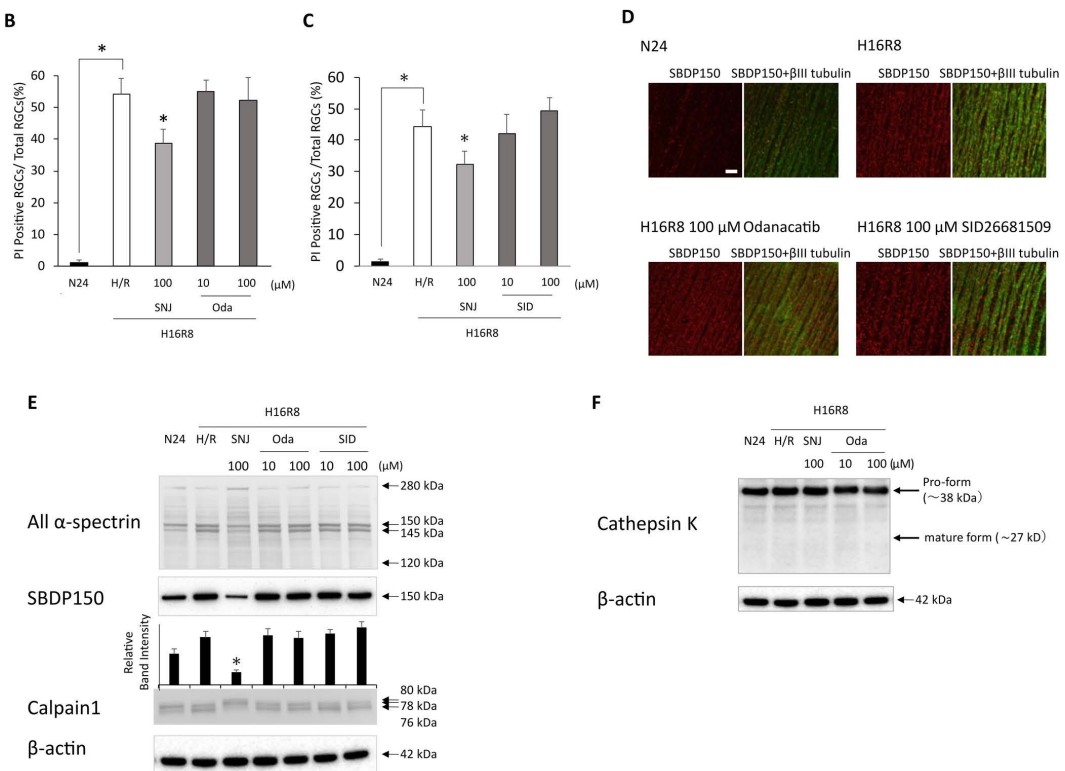

**A**

IC50 (nM)

| | Calpain 1 | Calpain 2 | Cathepsin K | Cathepsin B | Cathepsin L |
|---|---|---|---|---|---|
| **Odanacatib** (Cathepsin K inhibitor) | 15880 | No effect | 0.28 | 167 | 5127 |
| **SID26681509** (Cathepsin L inhibitor) | 80969 | 93183 | 1522 | 495 | 60 |

**Fig 1. The role of Cathepsin inhibitors in RGC damage in hypoxic retina.** (A) The IC50 values of cathepsin inhibitors against cathepsins and calpains in vitro. (B and C) Bar graphs display the percentage of PI-positive RGCs in flat-mount samples from hypoxic retina, indicating disruption of membrane integrity. Treatment with neither Odanacatib nor SID26681509 reduced the percentage of PI-positive RGCs, whereas the positive control SNJ-1945 significantly decreased the percentage. Data are means ± SD (n = 3). *P < 0.05 (Dunnett's test). Labels: N24, vehicle under normoxia for 24 hr; H16R8, vehicle under 16 hr hypoxia/8 hr reoxygenation; SNJ-1945, H16R8 with 100 μM SNJ-1945; Oda, H16R8 with 10 or 100 μM Odanacatib (B); SID, H16R8 with 10 or 100 μM SID26681509 (C). (D) Calpain activation in β-III tubulin-positive NFL from monkey retinal explants cultured under hypoxia/reoxygenation. Confocal microscopy of flat-mounts reveals calpain-specific SBDP150 immunoreactivity (red, left columns) colocalized with the neuronal marker β-III tubulin (green, right columns). Treatment with cathepsin inhibitors did not attenuate calpain activation. Scale bar, 50 μm. Representative images from 3 independent experiments are shown. (E) Immunoblots of monkey retina explants cultured under hypoxia/reoxygenation showing calpain 1 activation and proteolysis of α-spectrin. Treatment with Odanacatib or SID26681509 did not mitigate the activation of calpain 1 or proteolysis of α-spectrin, whereas the positive control SNJ-1945 effectively ameliorated these changes. Arrows indicate each target protein and their cleaved fragments. β-actin was used as an endogenous loading control. Representative images from 3 independent experiments are shown. (F) Immunoblot analysis of cathepsin K in monkey retina explants cultured under hypoxia/reoxygenation. The pro-form of cathepsin K was detected; however, no conversion to the mature form was observed during hypoxia/reoxygenation. Arrows indicate pro-form and mature-form of cathepsin K. β-actin was used as an endogenous loading control. Representative images from 3 independent experiments are shown.

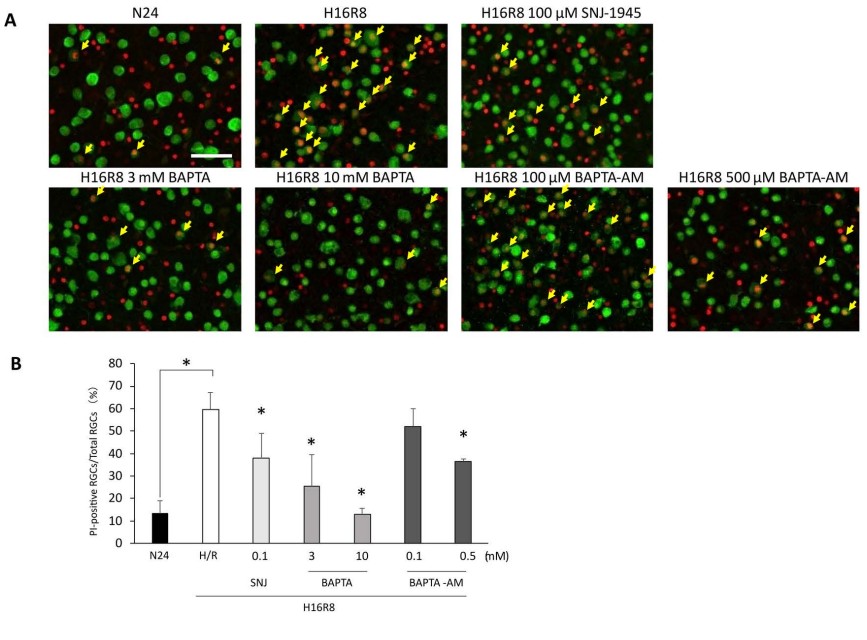

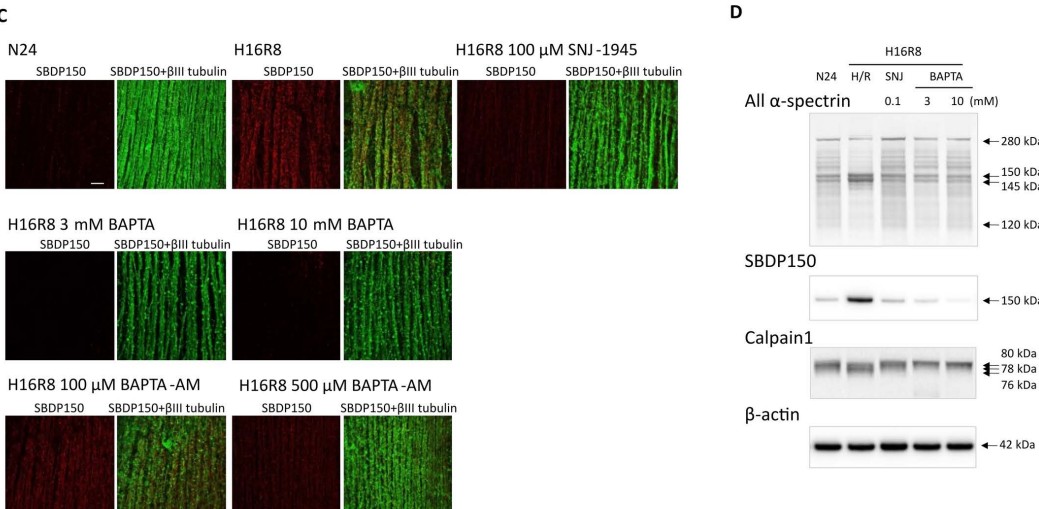

**Fig 2. Calcium ions are essential in RGC damage in hypoxic retina.** (A) Disruption of membrane integrity in RGCs under hypoxia/reoxygenation. Confocal microscopy of flat-mounted retina reveals colocalization of PI staining (red) with the RGC marker (green). Treatment with calcium chelators reduced the number of PI-positive RGCs, indicated by yellow arrows. The positive control SNJ-1945 also exhibited efficacy. Scale bar, 50 μm. Representative images from 3 independent experiments are shown. Labels: N24, vehicle under normoxia for 24 hr; H16R8, vehicle under 16 hr hypoxia/8 hr reoxygenation; SNJ-1945, H16R8 with 100 μM SNJ-1945; BAPTA, H16R8 with 3 or 10 mM BAPTA; BAPTA-AM, H16R8 with 0.1 or 0.5 mM BAPTA-AM. (B) Bar graphs display the percentage of PI-positive RGCs in flat-mount samples from hypoxic retina, indicating membrane integrity disruption. Treatment with BAPTA or BAPTA-AM reduced the percentage of PI-positive RGCs. The positive control SNJ-1945 also demonstrated efficacy in reducing PI-positive RGCs. Data are means ± SD (n = 3). *P < 0.05 (Dunnett's test). (C) Calpain activation in β-III tubulin-positive NFL from monkey retinal explants cultured under hypoxia/reoxygenation. Confocal microscopy of flat-mounts reveals calpain-specific SBDP150 immunoreactivity (red, left columns) colocalized with the neuronal marker β-III tubulin (green, right columns). Treatment with BAPTA or BAPTA-AM attenuated calpain activation. The positive control SNJ-1945 also demonstrated efficacy in reducing calpain activation. Scale bar, 50 μm. Representative images from 3 independent experiments are shown. (D) Immunoblots of monkey retina explants cultured under hypoxia/reoxygenation showing calpain 1 activation and proteolysis of α-spectrin. Treatment with BAPTA or BAPTA-AM mitigated the activation of calpain 1 or proteolysis of α-spectrin. The positive control SNJ-1945 also effectively ameliorated their changes. Arrows indicate each target protein and their cleaved fragments. β-actin was used as an endogenous loading control. Representative images from 3 independent experiments are shown.

## Other calpain inhibitors also suppressed RGC damage under hypoxia/reoxygenation

The IC50 values of each calpain inhibitor did not differ between human erythrocyte calpain 1 and porcine kidney calpain 2, suggesting that the inhibitors were unable to specifically discriminate between activated calpains (Fig 3A). The calpain inhibitors were subsequently tested using retinal cell lysate, which contained both calpains 1 and 2 activated by the addition of calcium. The immunoblotting results demonstrated that the addition of calcium led to an increase in calpain-specific SBDP150, which was attenuated by all calpain inhibitors, each exhibiting different levels of effectiveness. Calcium also induced the activation of calpains, as evidenced by the appearance of autolytic forms of calpain 1 at 78 kDa and 76 kDa [21], and the degradation of calpain 2 at 43 kDa, which appeared after calpain activation. SNJ-1945 and ABT-957 inhibited the activation of both calpain 1 and calpain 2 in a dose-dependent manner (Fig 3B). PD-151746 and C2I also inhibited calpain activation, but not as strongly as SNJ-1945 and ABT-957 (Fig 3B). These results showed that the four calpain inhibitors were not able to distinguish between calpain 1 and calpain 2 activation in our retinal model.

In hypoxic retinal explant model, these inhibitors reduced the numbers of PI-positive RGC cells (Figs 3C yellow arrows & 3D) and accumulation of SBDP150 in NFL (Fig 3E). Immunoblotting also showed that all inhibitors mitigated the activation of calpain 1 and the breakdown of α-spectrin (Fig 3F).

## Discussion

The present study offers two new insights, as follows: (1) Calcium-independent cathepsin is not the primary contributor to RGC damage in hypoxia-treated monkey retina. (2) Elevated cytosolic calcium originating from extracellular sources is crucial for activating calpains 1 and 2 and subsequent RGC damage in the hypoxic retina.

Our previous studies demonstrated that calpain activations lead to proteolysis of α-spectrin, resulting in damage to RGCs in monkey and human retinal explants subjected to hypoxia/reoxygenation [8,9]. Ischemia causes the translocation of calpains to the membranes of lysosomes, where cathepsins are typically located [22]. Activated calpain disrupts lysosomal membranes [23,24], leading to the subsequent release of cathepsins into the cytoplasm [25]. For instance, in Alzheimer's neuronal cells, calpain over-activation ruptures the Hsp70.1-mediated lysosomal membrane, resulting in cell death [26]. Additionally, calpain 1 is known to cleave lysosomal-associated membrane protein 2 (Lamp2), contributing to cell death [27]. The released cathepsins from the lysosome may proteolyze cytosolic proteins, acting as waste proteins within the lysosome. In the present experiment, odanacatib, a strong inhibitor of cathepsin K with moderate inhibition of cathepsin B, and the cathepsin L inhibitor SID did not prevent RGC damage under hypoxia/reoxygenation conditions in monkey explant cultures (Figs 1A, B, C).

### Patterns of α-spectrin degradation as evidence for the proteases responsible

Spectrin consisting of α-and β-subunits plays a key role in membrane integrity, cell shape, and cell-cell interactions. Spectrin subunits are degraded by several proteases, and α-spectrin breakdown pattern can be used to identify responsible proteases. Calpain proteolyzes α-spectrin to produce SBDP-150 and -145, with N-terminal sequences GMMPRD and SAHEVQ, respectively [28]. Caspase-3 generates SBDP-150 and -120, with N-terminal sequences SKTASP and SVEALI, respectively [29]. Although lysosomal proteases are known to degrade α-spectrin [30], the specific cleavage sites have not been reported. However, degraded α-spectrin produced by lysosomal exopeptidase cathepsins may appear at different positions on immunoblots compared to α-spectrin that has been proteolyzed by endopeptidases calpain and caspase-3, as calpain and caspase-3 generate relatively large and long-lived fragments. In the present experiment, increased SBDP-150 and -145 were observed as the only major proteolytic fragments in the retina treated with hypoxia (Fig 1E). Calpain inhibitors reduced the level of SBDP-150 and −145, while cathepsin inhibitors did not have any effect. A calpain-specific SBDP-150 also appeared, but the caspase-preferred SBDP-120 was not detected. Cathepsin L is an activator of caspase-3, which produces SBDP-120 [20]. This finding regarding calpain and caspase 3 is consistent with the previous report [8,9].

**Fig 3. Other calpain inhibitors in RGC damage in hypoxic retina.** (A) The IC50 of calpain inhibitors against calpains 1 and 2 in vitro.(B) Immunoblots of lysates from non-cultured monkey retina incubated for 2 hr with $CaCl_2$ showing autolysis of calpains 1 and 2, and proteolysis of α-spectrin. All calpain inhibitors mitigated the autolysis of calpains and proteolysis of α-spectrin. Arrows indicate each target protein and their cleaved fragments. β-actin was used as an endogenous loading control. Representative images from 3 independent experiments are shown. Labels: Veh, vehicle without $CaCl_2$; $Ca^{2+}$; vehicle with $CaCl_2$ (+ $Ca^{2+}$); SNJ, $Ca^{2+}$ with SNJ-1945; PD, $Ca^{2+}$ with PD-151746; ABT, $Ca^{2+}$ with ABT-957; C2I, $Ca^{2+}$ with C2I; 1, 1 μM; 10, 10 μM; 100, 100 μM. (C) Disruption of membrane integrity in RGCs under hypoxia/reoxygenation. Confocal microscopy of flat-mounted retina reveals colocalization of PI staining (red) with the RGC marker (green). All calpain inhibitors reduced the number of PI-positive RGCs, indicated by yellow arrows. Scale bar, 50 μm. Representative images from 3 independent experiments are shown. Labels: N24, vehicle under normoxia for 24 hr; H16R8, vehicle under 16 hr hypoxia/8 hr reoxygenation; SNJ, H16R8 with 100 μM SNJ-1945; PD-151746, H16R8 with 100 μM PD-151746; ABT-957, H16R8 with 100 μM ABT-957; C2I, H16R8 with 100 μM C2I. (D) Bar graph shows the percentage of PI-positive RGCs in the flat-mount samples from hypoxic retina, indicating disruption of membrane integrity. All calpain inhibitors decreased the percentage of PI-positive RGCs, exhibiting varying degrees of efficacy. Data are means ± SD (n = 3). *P < 0.05 (Dunnett's test). (E) Calpain activation in β-III tubulin-positive NFL from monkey retinal explants cultured under hypoxia/reoxygenation. Confocal microscopy of flat-mounts reveals calpain-specific SBDP150 immunoreactivity (red, left columns) colocalized with the neuronal marker β-III tubulin (green, right columns). All calpain inhibitors attenuated calpain activation. Scale bar, 50 μm. Representative images from 3

independent experiments are shown. (F) Immunoblots of monkey retina explants cultured under hypoxia/reperfusion showing calpain 1 activation and proteolysis of α-spectrin. All calpain inhibitors mitigated the activation of calpain 1 or proteolysis of α-spectrin. Arrows indicate each target protein and their cleaved fragments. β-actin was used as an endogenous loading control. Representative images from 3 independent experiments are shown.

## Active conversions as evidence for the role of responsible proteases

Active forms of proteases provide additional evidence for identifying the proteases involved. Cysteine proteases are expressed as inactive pro-forms, which become catalytically active through cleavage of the N-terminus [31]. In the present study, pro-form of cathepsin K detected in the monkey retina did not convert to active form under hypoxia conditions (Fig 1F). In contrast, the autocatalytic active form of calpain 1 [21] was observed in the hypoxic retina. This autolysis was inhibited by calpain inhibitor (Fig 1E).

## Calcium dependency as evidence for the proteases involved

The higher concentration of the calcium chelator, BAPTA almost completely inhibited RGC damage induced by hypoxia (Fig 2B). This strongly suggests that calcium is a crucial factor in RGC damage during hypoxia/reoxygenation. It also indicates that the influx of cytosolic calcium originates from the extracellular space rather than from organelles, as BAPTA is cell impermeable. BAPTA-AM, which can be hydrolyzed by esterase, penetrates the cell membrane, and chelates intracellular calcium [32]. However, BAPTA-AM only partially protected against RGC damage (Fig 2B), even at a higher concentration in our retinal explant compared to the concentration typically used in cell culture [33]. The concentration of BAPTA (hydrolyzed BAPTA-AM) may not be sufficient to chelate the increased cytosol calcium in our retinal explants, even though concentrations of BAPTA-AM as high as those soluble in the culture medium have been used.

Thus, the three pieces of data mentioned above suggest that calcium-dependent calpain is involved in proteolysis of α-spectrin and subsequent cell damage, while calcium-independent cathepsins and caspase-3 are not.

## Calcium-dependent proteolysis as a primary cause of RGC damage

Our previous report on human and monkey retinal explant cultures demonstrated that calpain inhibitor SNJ-1945 mitigated RGC cell damage induced by hypoxia/reoxygenation [8,9]. To confirm the involvement of calpain, several characteristic inhibitors were used in this experiment. SNJ-1945 and ABT-957 equally inhibited purified human calpain 1 and porcine calpain 2 as expected; SNJ-1945 demonstrated a slightly stronger effect than ABT-957 (Fig 3A). Unexpectedly, the calpain 1 specific inhibitor PD-151746 and the calpain 2 specific inhibitor C2I both equally inhibited calpain 1 and 2. C2I targets the catalytic triad in cysteine protease, as well as SNJ-1945 and ABT-957 [17]. PD-151746 targets the calcium binding site in C-terminal EF hand domain [16]. These targets, which include both calpains 1 and 2, may pose challenges in differentiating their protease specificity. An alternative explanation for this discrepancy could be the use of different sources of calpains in experiments. In retinal lysate, all inhibitors inhibited autolysis of calpains 1 and 2, and proteolysis of α-spectrin in the following order: SNJ-1945 > ABT-957 = C2I > PD-151746 (Figs 2D and 3F). All inhibitors reduced the production of SBDPs and cell damage in hypoxic retinal explants (Figs 3C, D, E, F). The specific contribution of calpain 1 and 2 to RGC damage could not be clarified in the present experiment due to the absence of isozyme-specific inhibition. Calpain 1 is preferably located in the nerve fiber layer (NFL) and the ganglion cell layer (GCL), while calpain 2 is distributed throughout the retina [9]. SNJ-1945 inhibited the autolysis of calpain 1 but did not affect that of calpain 2, as observed in immunoblotting of hypoxic retinal explants [8]. This discrepancy is thought to be attributed to the localization of calpain 2 in the inner retina, where SNJ-1945 could not penetrate effectively, despite being specifically designed to enhance membrane permeability [34]. In the present experiment, all inhibitors reduced the autolysis of calpain 1 in hypoxic retinal explants (Fig 3F), indicating their ability to penetrate RGCs. Immunoblotting of calpain 2 was not performed in this experiment based on previous findings. However, all inhibitors similarly inhibited both calpains 1 and 2 in retinal lysates, and

co-localization in RGCs from monkey retina suggested that the inhibitors could target calpain 2 in the same manner as calpain 1.

All inhibitors reduced RGC damage in hypoxic retinal explants (Fig 3D), but their effects were only partial and weaker compared to the calcium chelator BAPTA (Fig 2B). Even the most effective inhibitor, SNJ-1945, which nearly completely inhibited calpain-induced proteolysis similar to BAPTA (Figs 2D and 3F), did not match BAPTA's overall protective effect. The abnormality of the mitochondrial membrane has been documented to cause ATP depletion, ultimately resulting in necrosis [4]. Furthermore calcium/calmodulin-dependent protein phosphatase, calcineurin, contributes to neuronal damage in the ischemic rat retina [35]. Although speculative, additional calcium-dependent mechanisms, which remain unidentified, may also contribute to the RGC damage.

We hypothesized the underlying mechanism of RGC damage in monkey explant culture under hypoxia/reoxygenation conditions as follows: 1) calcium influx from the extracellular space into the cytosol, 2) an increase in cytosolic calcium, which subsequently activates both calpain 1 and calpain 2, leading to proteolysis and cellular damage.

To validate our hypothesis, further studies would be required, as our current experiment has limitations by using a restricted number of inhibitors and did not include direct measurement of cytosolic calcium levels and active calpain 2 in RGCs. In addition, only cathepsin K was analyzed for its activation. However, once specific cleavage sites for cathepsins are identified, additional evidence against cathepsin activation may be obtained.

## Supporting information

**S1 Fig. Original blot images of Figs 1E, 1F, 2D, 3B, 3F.**
(PDF)

## Acknowledgments

We thank Ms. Miho Makita (Senju Pharmaceutical, Co. Ltd., Laboratory of Ocular Sciences, Kobe, Japan) and Ms. Samantha Supnet (Senju Pharmaceutical, Co. Ltd., Laboratory of Ocular Sciences, Portland, Oregon) for their experimental support.

## Author contributions

**Conceptualization:** Mitsuyoshi Azuma, Emi Nakajima, Momoko Otsugu-Kobayashi, Takatoshi Uchida, Kana Orihara, Thomas R. Shearer.

**Data curation:** Mitsuyoshi Azuma, Emi Nakajima, Momoko Otsugu-Kobayashi, Takatoshi Uchida, Kana Orihara.

**Formal analysis:** Mitsuyoshi Azuma, Momoko Otsugu-Kobayashi, Takatoshi Uchida, Kana Orihara.

**Funding acquisition:** Mitsuyoshi Azuma.

**Investigation:** Emi Nakajima, Momoko Otsugu-Kobayashi, Takatoshi Uchida, Kana Orihara, Thomas R. Shearer.

**Methodology:** Mitsuyoshi Azuma, Emi Nakajima, Momoko Otsugu-Kobayashi, Takatoshi Uchida, Kana Orihara.

**Project administration:** Mitsuyoshi Azuma, Emi Nakajima, Momoko Otsugu-Kobayashi, Takatoshi Uchida, Kana Orihara, Thomas R. Shearer.

**Resources:** Mitsuyoshi Azuma, Momoko Otsugu-Kobayashi, Takatoshi Uchida, Kana Orihara.

**Supervision:** Mitsuyoshi Azuma, Emi Nakajima, Takatoshi Uchida, Kana Orihara, Thomas R. Shearer.

**Validation:** Mitsuyoshi Azuma, Emi Nakajima, Momoko Otsugu-Kobayashi, Takatoshi Uchida, Kana Orihara, Thomas R. Shearer.

**Visualization:** Momoko Otsugu-Kobayashi, Takatoshi Uchida, Kana Orihara.

**Writing – original draft:** Emi Nakajima, Momoko Otsugu-Kobayashi, Kana Orihara, Thomas R. Shearer.

**Writing – review & editing:** Mitsuyoshi Azuma, Emi Nakajima, Momoko Otsugu-Kobayashi, Takatoshi Uchida, Thomas R. Shearer.

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
