## [Decision Letter · Decision Letter 0]

PONE-D-25-08717Calcium-dependent pathway as a primary cause of hypoxic RGC damage in monkey retinal explantsPLOS ONE

Dear Dr. Azuma,

Thank you for submitting your manuscript to PLOS ONE. After careful consideration, we feel that it has merit but does not fully meet PLOS ONE’s publication criteria as it currently stands. Therefore, we invite you to submit a revised version of the manuscript that addresses the points raised during the review process.Please notice that corrections were suggested to the manuscript. The authors should follow the suggestions or provide a rebuttal letter.

We look forward to receiving your revised manuscript.

Kind regards,

Alexandre Hiroaki Kihara, Ph.D.

Academic Editor

PLOS ONE

Journal Requirements:

2. In your Methods section, please indicate the source of the porcine kidneys (line 107). If animals were sacrificed for the specific purposes of this research, please provide the full name of the relevant ethics committee that approved the work, the associated permit number(s), and additional information on the animal research and ensure you have included details on (1) methods of sacrifice, (31) methods of anesthesia and/or analgesia, and (3) efforts to alleviate suffering.

[This research is fully funded by Senju Pharmaceutical Co., Ltd. in Japan.]. 

Please respond by return e-mail so that we can amend your financial disclosure and competing interests on your behalf.

[This research was supported in part by National Institutes of Health Grant P51OD011092 to the Oregon National Primate Research Center.

We thank Ms. Miho Makita (Senju Pharmaceutical, Co. Ltd., Laboratory of Ocular Sciences, Kobe, Japan) and Ms. Samantha Supnet (Senju Pharmaceutical, Co. Ltd., Laboratory of Ocular Sciences, Portland, Oregon) for their experimental support.]

[This research is fully funded by Senju Pharmaceutical Co., Ltd. in Japan.]

6. Thank you for stating the following in the Competing Interests section:

[I have read the journal's policy and the authors of this manuscript have the following competing interests: Dr. Shearer has significant financial interests (research contract and/or consulting fee) in Senju Pharmaceutical Co., Ltd., and Drs. Azuma, Nakajima and Uchida are employees of Senju Pharmaceutical Co., Ltd. a company that may have a commercial interest in the results of this research and technology. These potential conflicts of interest have been reviewed and managed by OHSU Conflict of Interest in Research Committee.]. 

We note that you received funding from a commercial source: [Senju Pharmaceutical Co., Ltd.]

7. PLOS ONE now requires that authors provide the original uncropped and unadjusted images underlying all blot or gel results reported in a submission’s figures or Supporting Information files. This policy and the journal’s other requirements for blot/gel reporting and figure preparation are described in detail at https://journals.plos.org/plosone/s/figures#loc-blot-and-gel-reporting-requirements and https://journals.plos.org/plosone/s/figures#loc-preparing-figures-from-image-files. When you submit your revised manuscript, please ensure that your figures adhere fully to these guidelines and provide the original underlying images for all blot or gel data reported in your submission. See the following link for instructions on providing the original image data: https://journals.plos.org/plosone/s/figures#loc-original-images-for-blots-and-gels.  

Reviewers' comments:

Reviewer's Responses to Questions

**Comments to the Author**

1. Is the manuscript technically sound, and do the data support the conclusions?

Reviewer #1: Yes

2. Has the statistical analysis been performed appropriately and rigorously? 

Reviewer #1: Yes

3. Have the authors made all data underlying the findings in their manuscript fully available?

Reviewer #1: Yes

4. Is the manuscript presented in an intelligible fashion and written in standard English?

Reviewer #1: Yes

5. Review Comments to the Author

Reviewer #1: In the manuscript entitled ”Calcium-dependent pathway as a primary cause of hypoxic RGC damage in monkey retinal explants”, the authors primarily focused on retinal ganglion cell (RGC) damage, investigating both calcium-dependent and calcium-independent mechanisms involving calpain and cathepsins. This study builds upon the authors' previous work, where they observed the involvement of calpains. The current study focuses on whether RGC damage is mediated by calcium-dependent calpains, rather than by calcium-independent cathepsins.

Comments:

Given that the authors are investigating the involvement of calcium-independent cathepsins, could they clarify why experiments to measure the expression or presence of cathepsins in retinal lysates or flat mounts, as well as any associated quantitative analyses, were not performed in this study?

Can the authors kindly provide the specific reference numbers for references 16-19 within the text, as they appear to be missing?

Considering that there are 11 types of cysteine cathepsins present, could the authors kindly elaborate on why they specifically chose to focus on cathepsins K, B, and L? Was there a particular rationale behind this selection?

Line 255: Since the labeling in Figure 1 is not very clear, but assuming that SNJ treatment mitigated RGC death, could the authors please clarify why the reference is included there? What specific information or support do the authors intend to convey with this reference? It appears that references 21 and 22 are identical.

As mentioned in the Materials and Methods section, the authors did not provide any evidence or representative images demonstrating the activity of cathepsins and their effects on the substrate. Instead, the focus seems to be primarily on calpains. Could the authors clarify why the activity of cathepsins was not further investigated or visualized in this study?

Line 266: Although the labeling is not very clear, assuming that the third lane represents the treatment with SNJ, as indicated by the authors, there doesn't appear to be a decrease in SBDP150 levels. Could the authors kindly explain this discrepancy? Additionally, was any quantitative analysis performed to assess the changes in SBDP150 levels following treatments?

In Figure 2a, the panel for H16R8 500 μM shows a decrease in RGC numbers. Although the pretreatment was administered 2 hours prior to hypoxic induction, could the authors discuss what might have caused the observed decrease in RGC numbers?

Could the authors provide a label for calpain 2 in Figure 3?

6. PLOS authors have the option to publish the peer review history of their article (what does this mean? ). If published, this will include your full peer review and any attached files.

**Do you want your identity to be public for this peer review?** For information about this choice, including consent withdrawal, please see our Privacy Policy .

Reviewer #1: No

---

## [Author Response · Author response to Decision Letter 1]

10 Jun 2025

The answers to the journal requirements:

1. The manuscript was revised to meet PLOS ONE formatting requirements.

2. The source of porcine kidneys (line 107) was added as follows.

“The porcine kidney was obtained from DART Co., Ltd. as part of a program aimed at the effective utilization of livestock by-products in Hokkaido, Japan. The procurement process followed ethical guidelines from Livestock Industry Bureau MAFF-J (ministry of agriculture, forestry and fisheries of Japan) for farm animal use, ensuring compliance with relevant regulations.”

3. There are no grant numbers for each research project at Senju Pharmaceutical. Please change the financial disclosure as below in “4”.

4. Please change the statement of Senju Pharmaceutical to “The research was supported by funding from Senju Pharmaceutical Co., Ltd. Senju Pharmaceutical Co., Ltd. played a role in the design of the study and the the interpretation of the data. The authors declare that there is a potential conflict of interest because the research results may benefit Senju Pharmaceutical Co., Ltd.’s products.”

5. “This research was supported in part by National Institutes of Health Grant P51OD011092 to the Oregon National Primate Research Center.” was removed from the acknowledgement section. Please add this sentence to our funding information.

Should we write about this grant also in the financial disclosure above? If so, “the funder had no role in study design, data collection and analysis, decision to publish, or preparation of the manuscript” for this.

6. The amended competing interests reads “Dr. Shearer was a paid consultant for Senju Pharmaceutical Co., Ltd., a company that may have a commercial interest in the results of this research and technology. Drs. Azuma, Nakajima, and Uchida are employees of Senju Pharmaceutical Co., Ltd., a company that may have a commercial interest in the results of this research and technology. These potential conflicts of interest were reviewed, and a management plan approved by the OHSU Conflict of Interest in Research Committee was implemented.”

7. Original uncropped and unadjusted images were compiled and uploaded along with the revisions as supporting information (file name: original blot images.pdf)

8. The mistakes in the reference list were corrected and included in the Response to Reviewers letter.

The answers to reviewer's comments:

Given that the authors are investigating the involvement of calcium-independent cathepsins, could they clarify why experiments to measure the expression or presence of cathepsins in retinal lysates or flat mounts, as well as any associated quantitative analyses, were not performed in this study?

As noted in the Discussion, specific cleavage sites of α-spectrin by cathepsins have not been identified. Therefore, unlike calpain activation, α -spectrin cannot be used as a reliable marker for cathepsin activity. This limitation has now been stated in the revised Discussion.

Instead, we analyzed the conversion of pro-cathepsin K to its mature form as an indicator of cathepsin activation. Our results showed that hypoxia/reoxygenation did not induce the conversion of pro-cathepsin K to its mature form, in contrast to the activation observed for calpain 1. These findings have presented in Figure 1F.

However, it should be noted that only cathepsin K was examined in this study, although 11 cathepsins have been reported in the literature. This limitation has also been added to the revised Discussion.

Can the authors kindly provide the specific reference numbers for references 16-19 within the text, as they appear to be missing?

References 16 and 17 were used in the previous version, and we forgot to remove them when we revised the manuscript for submission. They are removed from the refence section, and the rest of the references were re-numbered. References 18 and 19 were re-numbered to references 14 and 15, respectively. We also found that references 23 and 24 were identical. Reference 24 was removed.

Considering that there are 11 types of cysteine cathepsins present, could the authors kindly elaborate on why they specifically chose to focus on cathepsins K, B, and L? Was there a particular rationale behind this selection?

The rationale for focusing on cathepsins K, B, and L lies in their well-documented involvement in cell death pathways and their early release following lysosomal membrane permeabilization. This explanation has now been included in the Materials and Methods section, along with appropriate citation (reference 13).

Line 255: Since the labeling in Figure 1 is not very clear, but assuming that SNJ treatment mitigated RGC death, could the authors please clarify why the reference is included there? What specific information or support do the authors intend to convey with this reference? It appears that references 21 and 22 are identical.

Figure 1B and 1C have been modified with clearer labeling. We hope this clarifies the mitigation of RGC death following SNJ-1945 treatment. Reference 21 was included because the effect of SNJ-1945 shown in Figure 1 was reproduced based on the findings reported in that reference. This has been clarified in line 258. As suggested by the reviewer, Reference 22 has been removed, as it was identical to Reference 21.

As mentioned in the Materials and Methods section, the authors did not provide any evidence or representative images demonstrating the activity of cathepsins and their effects on the substrate. Instead, the focus seems to be primarily on calpains. Could the authors clarify why the activity of cathepsins was not further investigated or visualized in this study?

As mentioned in the Materials and Methods section, the transformation of pro-cathepsin K to its mature form was used as a marker of cathepsin activation. This has now been clearly described in the revised manuscript. However, no such conversion was observed under our experimental conditions (Figure 1F), suggesting that cathepsin K was not activated.

Further investigation into cathepsin activity was not conducted in this study due to insufficient information, such as the lack of identified specific cleavage sites for cathepsins.

Instead, cathepsin inhibitors were used to gain partial insight into their potential role. However, these inhibitors did not suppress any of the changes induced by hypoxia/reoxygenation.

Taken together, these findings suggest that we were only able to partially evaluate cathepsin activity under the current conditions. We recognize the importance of further investigation, which will be considered in future studies. As explained in our responses to the previous comments, these points have now been incorporated into the revised manuscript.

Line 266: Although the labeling is not very clear, assuming that the third lane represents the treatment with SNJ, as indicated by the authors, there doesn't appear to be a decrease in SBDP150 levels. Could the authors kindly explain this discrepancy? Additionally, was any quantitative analysis performed to assess the changes in SBDP150 levels following treatments?

Labels for Fig. 1E and 1F have been modified with clearer labeling, and unnecessary lane 1 has been removed. We hope this clarifies the decrease in SBDP150 levels in the third lane of revised Figure 1E. As suggested by the reviewer, a graph showing the densitometric analysis has been added the blot image in Figure 1E to provide quantitative support for the observed changes.

In Figure 2a, the panel for H16R8 500 μM shows a decrease in RGC numbers. Although the pretreatment was administered 2 hours prior to hypoxic induction, could the authors discuss what might have caused the observed decrease in RGC numbers?

We would like to clarify that Figure 2a shows PI-positive RGCs, which represent damaged or dying cells. As the reviewer noted, treatment with 500 µM BAPTA-AM resulted in a decrease in the number of PI-positive RGCs following H16R8 exposure. This indicates that BAPTA-AM, when administered 2 hours prior to hypoxic induction, effectively mitigated RGC damage. Therefore, the observed decrease reflects a positive effect, rather than a loss of total RGCs.

Could the authors provide a label for calpain 2 in Figure 3?

Labels for Calpain 2 intact band and the cleaved band are now added in Figure 3B.

---

## [Editor Report · Decision Letter 1]

Calcium-dependent pathway as a primary cause of hypoxic RGC damage in monkey retinal explants

PONE-D-25-08717R1

Dear Dr. Azuma,

We’re pleased to inform you that your manuscript has been judged scientifically suitable for publication and will be formally accepted for publication once it meets all outstanding technical requirements.

Kind regards,

Alexandre Hiroaki Kihara, Ph.D.

Academic Editor

PLOS ONE
---

## [Editor Report · Acceptance letter]

PONE-D-25-08717R1

PLOS ONE

Dear Dr. Azuma,

I'm pleased to inform you that your manuscript has been deemed suitable for publication in PLOS ONE. Congratulations! Your manuscript is now being handed over to our production team.

Kind regards,

on behalf of

Dr. Alexandre Hiroaki Kihara

Academic Editor

PLOS ONE